# Loads of Coliforms and Fecal Coliforms and Characterization of Thermotolerant *Escherichia coli* in Fresh Raw Milk Cheese

**DOI:** 10.3390/foods11030332

**Published:** 2022-01-25

**Authors:** Ahmed M. Hammad, Amira Eltahan, Hamdy A. Hassan, Nasser H. Abbas, Heba Hussien, Tadashi Shimamoto

**Affiliations:** 1Department of Food Hygiene and Control, Faculty of Veterinary Medicine, University of Sadat City, Sadat City 32897, Egypt; hammad@vet.usc.edu.eg (A.M.H.); amira.eltah@gmail.com (A.E.); hebashelaby2013@yahoo.com (H.H.); 2Department of Environmental Biotechnology, Genetic Engineering and Biotechnology Research Institute, University of Sadat City, Sadat City 32897, Egypt; hamdy.hassan@gebri.usc.edu.eg (H.A.H.); nasser.abbas@gebri.usc.edu.eg (N.H.A.); 3Department of Biological Science, Faculty of Science and Humanity Studies at Al-Quwayiyah, Riyadh 11961, Saudi Arabia; 4Laboratory of Food Microbiology and Hygiene, Graduate School of Integrated Sciences for Life, Hiroshima University, Higashi-Hiroshima 739-8528, Japan

**Keywords:** thermotolerant coliforms, cheese, antibiotic resistance, virulence, Shiga toxin-producing *E. coli* (STEC)

## Abstract

The aim of this study was to assess the hygienic status of raw milk cheese and determine the trends of virulence and antimicrobial resistance in thermotolerant *Escherichia coli*. Two hundred samples of karish, a popular Egyptian fresh raw milk cheese, were analyzed for coliforms and fecal coliforms using a standard most probable number (MPN) technique. Overall, 85% of samples were unsuitable for consumption, as they exceeded Egyptian standards for coliforms (10 MPN/g), and 65% of samples exhibited coliforms at 44.5 °C. Of 150 recovered thermotolerant strains, 140 (93.3%) were identified as *E. coli*. Importantly, one Shiga toxin-producing *E. coli* (STEC) strain carrying a striking virulence pattern, *stx*1−, *stx*2+, *eae*−, was detected. Eleven strains (7.8%, 11/140) showed resistance to third-generation cephalosporins. Antibiotic resistance genes included *bla*_SHV_, *bla*_CTX-M_, *qnrS*, *tet*(A), and *tet*(B), which were present in 4.3%, 2.8%, 0.71%, 2.1%, and 0.71% of isolates, respectively. In conclusion, this study indicated that hygienic-sanitary failures occurred throughout the production process of most retail karish cheese. Furthermore, our findings emphasize the need for adopting third-generation cephalosporin-resistant *E. coli* as an indicator for monitoring antimicrobial resistance in raw milk cheese to identify the potential public health burden associated with its consumption.

## 1. Introduction

Cheese making is a primary industry worldwide. In general, cheese is still made on a relatively small scale. The traditional practices for preparing raw milk cheeses account for the rich diversity of cheeses available [1]. Several types of fresh raw milk cheeses are consumed worldwide, such as quark, chevre, ricotta, cottage, and queso fresco. Karish is the most popular fresh cheese in Egypt. Like quark and cottage cheese, karish cheese is manufactured without the addition of rennet. It is an ancient type of white, soft, lactic cheese made from curdled skim milk curd, “Laban Rayeb”, and neither acidifying agents nor lactic acid bacteria are added. However, lactic acid is created when lactose is fermented by the endogenous microflora of milk during the cheesemaking process, which is potent against some bacteria [2]. In general, bacterial contamination of raw milk during the milking process or from environmental sources (such as contaminated water or milking equipment) is a significant contributor to the microbial contamination of raw milk cheeses [3,4]. In processing plants, workers’ hands and cheese-contact surfaces have been recognized as primary sources for contamination with spoilage and pathogenic bacteria [2].

Coliforms are a group of Gram-negative, non-spore-forming, rod-shaped aerobes and facultative anaerobes that inhabit the intestinal tract of all vertebrates. They can ferment lactose when incubated at 35–37 °C, producing acid and gas [5]. The coliform bacteria that grow at 44.5 °C are referred to as fecal coliforms, including *Escherichia coli, Klebsiella pneumoniae*, *Enterobacter agglomerans*, *Enterobacter aerogenes*, *Enterobacter cloacae*, and *Citrobacter freundii* [6]. It is now clear that the ability of fecal coliforms to grow at high temperatures has little relationship with their fecal origin [7]. However, fecal coliforms are considered more directly associated with fecal contamination of mammals than other members of the coliform group [8]. Therefore, in the European Union, New Zealand, and Australia, the “fecal coliform” term has been substituted by what is considered a more suitable descriptor of this group of bacteria, “thermotolerant coliforms” (THC) [9]. Of note, despite misinterpretation of results of the coliform assay, current food and water quality regulations all over the world are still primarily based on coliform concentrations. They are a commonly used indicator of the hygienic quality of foods and water. It should be mentioned that the Egyptian standard of karish cheese lacks a limit for fecal coliform counts, and thus the acceptability of karish cheese is based on the count of coliforms [10].

*E. coli* is an extremely diverse species, including strains with little or no pathogenic potential and those that have acquired chromosomal or extra-chromosomal virulence operons to become highly infectious and virulent towards humans, animals, or both [11]. Pathotypes of diarrheagenic *E. coli* are grouped according to their virulence genes into enteropathogenic *E. coli* (EPEC), enterotoxigenic *E. coli* (ETEC), enteroinvasive *E. coli* (EIEC), diffuse adherent *E. coli* (DAEC), enteroaggregative *E. coli* (EAEC), adherent-invasive *E. coli* (AIEC), cell-detaching *E. coli* (CDEC), and Shiga toxin-producing *E. coli* (STEC) [12]. The latter pathotype includes a subgroup designated as enterohemorrhagic *E. coli* (EHEC), which is characterized by harboring the *eae* gene (encodes intimin, which is involved in attaching and effacing phenotype). The pathogenicity of STEC is mainly reliant on the development of one or two Shiga toxins (Stx1, Stx2), causing mild to severe symptoms of hemorrhagic colitis (HC). However, two major lethal syndromes, namely hemolytic uremic syndrome (HUS) and thrombotic thrombocytopenic purpura (TTP), have been reported as outcomes associated with STEC infection [13].

The evolution of resistance to clinically important antibiotics in foodborne bacteria, which can transmit antibiotic resistance determinants to human pathogens, is a critical public health concern. Notably, the propensity of *E. coli* to interchange genetic material with different bacteria makes them great candidates for understanding the reservoir of resistance genes in food of animal origin. One of the most prominent resistance mechanisms in *E. coli* that have been widely reported worldwide is the production of β-lactamases, particularly extended-spectrum β-lactamases (ESBL), which confer resistance to extended-spectrum cephalosporins such as cefotaxime, ceftriaxone, and ceftazidime [14]. Worryingly, ESBL-producing bacteria have shown co-resistance to several other classes of antimicrobials such as aminoglycosides, tetracyclines, quinolones, chloramphenicol, and sulfonamides. Consequently, third-generation cephalosporin-resistant *E. coli* was selected as an indicator to monitor antimicrobial resistance in food animals in the European Union [15]. However, there is a dearth of information about using such indicators for the surveillance of resistance to zoonotic antimicrobials in food of animal origin, particularly raw milk cheese. It is worth mentioning that harmonization of surveillance programs of antimicrobial resistance in humans, animals, and foods of animal origin all over the world, using known indicators, is important from a One Health perspective [16].

Of note, there are many studies on the phenotypic and molecular characteristics of coliforms and fecal coliforms in water and some dairy foods [6,17], but there are few reports on the phenotypic and molecular characteristics of this group in unpasteurized dairy products. Therefore, this study aimed to assess the hygienic status of raw milk cheese in Egypt, which could contribute to the development of public health protection monitoring and control activities for dairy industries.

## 2. Materials and Methods

### 2.1. Collection of Samples

A total of 200 karish cheese samples were collected from different small-scale supermarkets (60 samples), retail (100 samples), and dairy shops (40 samples) in the El-Menofia governorate of Egypt. All samples were produced not more than one day before the collection date and kept in the refrigerator for sale. Samples were collected in aseptic packages and transferred to our laboratory in an icebox. Upon arrival at the laboratory, they were maintained in the refrigerator at 4 °C and examined within 6 h of their arrival.

### 2.2. Fecal Coliform Analysis

Cheese samples (25 g) were diluted with 225 mL (dilution 1:10) of buffered peptone water (BPW) (Oxoid, Hampshire, UK) and homogenized in a stomacher. Seven consecutive dilutions (from 10^−2^ to 10^−8^) were performed. Coliform populations were estimated using the three-tube most-probable-number (MPN) method [18]. Briefly, one milliliter of the diluted cheese sample (10^−1^ to 10^−8^) was added to three lauryl sulfate tryptose broth tubes (9 mL) (Merck, Darmstadt, Germany) with inverted Durham tubes and incubated at 37 °C for 48 h [18]. All gas-positive samples were then inoculated into *Escherichia coli* broth (EC broth) (9 mL) (Oxoid, Hampshire, UK) and incubated at 44.5 °C for 48 h.

### 2.3. Bacterial Identification

The EC broth-positive samples were streaked onto eosin-methylene blue agar (EMB) (Oxoid, Hampshire, UK) and incubated at 35 °C for 24 h. Three colonies representing different morphological types on the EMB plates were subcultured for subsequent characterization. Isolated colonies were biochemically confirmed, as previously described [18].

### 2.4. Antimicrobial Susceptibility Testing

All isolates were characterized by antimicrobial susceptibility testing for 16 antimicrobials (Oxoid, Hampshire, UK) using a disc diffusion method in accordance with the Clinical and Laboratory Standards Institute (CLSI) guideline [19]. The selected antimicrobials are important in both clinical and agricultural settings and utilized to assess acquired resistance in *E. coli* and other Enterobacteriaceae [15,19]. They were as follows (µg): amoxicillin-clavulanic acid (20/10), ampicillin (10), aztreonam (30), cefepime (30), cefixime (5), cefotaxime (30), cefpirome (30), ceftazidime (30), ceftriaxone (30), ciprofloxacin (5), gentamicin (10), imipenem (10), nalidixic acid (30), streptomycin (10), sulfamethoxazole-trimethoprim (23/75), and tetracycline (30).

### 2.5. Screening and Confirmatory Tests for β-Lactamase and Extended-Spectrum β-Lactamase (ESBL) Production

All isolates were tested for β-lactamase production using nitrocefin (Oxoid, Hampshire, UK), as described previously [20]. Isolates showing β-lactamase production in the nitrocefin test were further screened for ESBL production by the double-disc synergy test (DDST). DDST was performed as a standard disc diffusion assay on Mueller Hinton agar (MHA) following CLSI recommendations [19]. All presumptive ESBL-producing isolates were subjected to phenotypic confirmation tests using ESBL clavulanic acid (CLA) combination discs [19]. Cefotaxime/clavulanic acid (CTX/CLA; 30/10 μg) and ceftazidime/clavulanic acid (CAZ/CLA; 30/10 μg) discs (Becton Dickinson, East Rutherford, NJ, USA) were used. ESBL production was defined as an increase in the diameter of the zone of inhibition of ≥5 mm for either ceftazidime or cefotaxime in combination with clavulanic acid compared with the diameter when tested alone [19].

### 2.6. Molecular Analysis

Total DNA was extracted using a rapid boiling procedure, as previously described [21]. Enterobacterial repetitive intergenic consensus (ERIC)-PCR [22] and susceptibility patterns were used for clone delineation. One representative per clone was included. *E. coli* isolates were assigned to phylogenetic groups, as described previously [23]. The presence of virulence and antibiotic resistance genes was identified by PCR in isolated strains. Primer sequences are listed in Appendix A.

### 2.7. Statistical Method

Box-and-whisker plots using log10-transformed data of MPN/g were produced to visually explore the dispersions and central tendencies of the data [24]. The boxes illustrate median values (horizontal line through the box) and the interquartile range; the whisker lines represent the lowest and highest values within 1.5 times the interquartile range. Violon plots were also produced to represent the frequency of samples [24]. Both box-and-whisker and violin plots were prepared using R studio [25].

### 2.8. Visualization of Genotypic and Phenotypic Patterns

The ComplexHeatmap (v2.6.2) R package [26] was used to plot a summary heatmap for the resistance or susceptibility to different antimicrobials and the virulence patterns of our STEC strain and STEC strains identified in 17 studies published in the last decade.

## 3. Results and Discussion

Coliforms were detected in 94% (188/200) of analyzed cheese samples. The majority of samples (85%, 170/200) were unsuitable for consumption as they exceeded the maximum Egyptian standard for coliforms (10 MPN/g) [10]. The visual distribution of the MPN/g of coliforms and fecal coliforms is presented in Figure 1. The coliform counts ranged from a minimum of 3 MPN/g to a maximum of 9.3 × 10^5^ MPN/g with a mean value of 1.5 × 10^5^. Of note, 75% of samples contained more than 1.1 × 10^3^ MPN/g (above the first quartile). However, a higher count of coliforms with a mean value of 2.9 × 10^7^ cfu/mL was previously reported in Brazil [27]. On the other hand, in the US, a considerable percentage of raw milk cheeses (42%) were contaminated with coliforms at a concentration of > 10 cfu/g [7]. Our data reveal how unsanitary manufacturing practices during the production and handling of raw milk cheese may jeopardize the product’s microbiological quality, the situation that was reported in both developed and developing countries.

On the other hand, 65% (130/200) of samples exhibited fecal coliforms at 44.5 °C. The load of contamination by fecal coliforms was within the range of 16 to 3.4 × 10^5^ MPN/g, with 75% of samples containing more than 6.4 × 10^3^ MPN/g (Figure 1). Notably, about half of the samples (47.7%, 62/130) had fecal coliforms with concentrations between 10^4^ and 10^5^ MPN/g. Similarly, Araujo et al. [28] reported a distinct distribution of fecal coliforms, with 93.3% of samples containing fecal coliforms from 10^5^ to 10^6^ MPN/g. Interestingly, in developing countries, the dissemination of food safety awareness material has been minimal, and this study underlines the need for better hygiene measures for preparing raw milk cheese. Enhanced hygiene practices, including pasteurization on dairy farms and at small factories, should be carried out to improve the quality and safety of raw milk cheese.

Thermotolerant *E. coli* is the primary bacterium in fecal coliforms and is considered the most precise indication of fecal contamination because it does not come from the environment but is only seen in high quantities in the feces of humans and warm-blooded animals [5,7]. In this study, 150 strains were isolated at 44.5 °C. Analysis of the population structure of isolated strains revealed that most isolates were thermotolerant *E. coli* (93.3%, 140/150) and *K. pneumoniae* (6.7%, 10/150). The high occurrence of thermotolerant *E. coli* in our samples revealed possible fecal contamination during milk collection and cheese processing. Regarding the limit of *E. coli*, unlike several international standards that set a permissible limit of 10–1000 *E. coli* in cheese samples [5], the Egyptian standard stated that karish cheese must be free from *E. coli* [10]. Given that *E. coli* was not detected in five cheese samples out of 130 samples containing fecal coliforms, thus 62.5% (125/200) of analyzed samples did not comply with the Egyptian standard. Higher occurrences of *E. coli* at percentages of 100% [27] and 97.7% [28] were detected from cheese samples in Brazil. On the other hand, lower occurrences of 40% [29] and 22% [7] were reported in Mexico and the US, respectively. It is worth mentioning that while *E. coli* and fecal coliforms are thought to be linked to fecal contamination, they do not provide a sufficient level of resolution to distinguish between human and animal transmission routes.

Phylogenetic analyses have shown that *E. coli* strains belong to five phylogenetic groups (A, B1, B2, D, and F) [23]. Most commensal strains belong to groups A and B1, whereas virulent extra-intestinal strains belong mainly to group B2 and, to a lesser extent, group D [30]. Interestingly, detecting a specific phylogenetic group or groups in food isolates may prove valuable as a risk assessment marker for food safety research. In previous studies, animal and human *E. coli* isolates exhibited significant shifts in phylogenetic distribution, with the former being mainly from phylogenetic groups B1, while the latter was predominantly from phylogenetic group A [31,32,33]. In this study, phylogenetic analysis revealed that 88.6% (124/140) of *E. coli* isolates belonged to group A, and some were from group B1, 11.4% (16/140), suggesting that indirect fecal contamination probably originated from the cheese makers because of poor hygienic practices during the production of karish cheese.

Analysis of the susceptibility status of THC isolates revealed that 60 (42.8%, 60/140) isolates were susceptible to all antibiotics tested, 47 (33.6%, 47/140) isolates were resistant to one antibiotic, and 20 (14.3%, 20/140) isolates were resistant to two antibiotics. Multiple resistance (to three or more antimicrobial agents) was observed in 13 isolates (9.3%, 13/140). The phenotypic and genotypic characteristics of 80 thermotolerant *E. coli* strains that showed resistance to at least one antimicrobial are presented in Figure 2. Overall, the highest rates of resistance were observed for ampicillin (30.7%, 43/140), cefixime (16.4%, 23/140), streptomycin (16.4%, 23/140), and tetracycline (12.8%, 18/140). Of note, none of the isolated strains were resistant to the fourth generation cephalosporins (cefepime and cefpirome), a carbapenem (imipenem), and monobactam (aztreonam). Visualization of the antimicrobial resistance profiles of isolated strains using a heatmap revealed three clusters (Figure 2). Cluster 1 and 2 included strains that were not resistant to any of the third-generation cephalosporins tested in this study. The majority of the strains (83.7%, 67/80) showed resistance to one or two classes of antimicrobials and were included in cluster 1. Cluster 2 included two strains, KC-9 and KC-42, which showed resistance to antimicrobials belonging to three or more classes of antimicrobials and thus classified as multidrug-resistant strains. Cluster 3, on the other hand, was made up exclusively of isolates that were resistant to one or more third-generation cephalosporins as well as one or more antimicrobial classes.

Interestingly, in the recent ranking of medically essential antimicrobials conducted by the World Health Organization (WHO), third-generation cephalosporins were classified as “highest priority critically important antimicrobials” [34]. Thus, a closer follow-up for the emergence and dissemination of resistance to this group of antibiotics in animal and food of animal origin is essential for public health. A striking finding in this study is that 11 strains (7.8%, 11/140), which made up cluster 3, showed resistance to two or three clinically important third-generation cephalosporins, including cefotaxime (6.4%, 9/140), ceftriaxone (5.7%, 8/140) and ceftazidime (7.1%, 10/140) (Figure 2). Worryingly, nine (6.4%, 9/140) strains were confirmed as ESBL-producers (KC-10, KC-18, KC-32, KC-40, KC-50, KC-53, KC-54, and KC-60), five (3.6%, 5/140) strains (KC-40, KC-50, KC-53, KC-60, and KC-28) showed the antibiotic resistance pattern, “CRO (ceftriaxone) + CTX (cefotaxime) + CAZ (ceftazidime)”, and three (2.1%, 3/140) strains (KC40, KC-53, and KC54) considered as multidrug-resistant strains because they showed co-resistance to at least three different classes of antimicrobials.

Notably, the discovery of a ceftriaxone-resistant strain, KC-28, that was not an ESBL producer supports Tamma and Humphries’ hypothesis [35] that non-susceptibility to third-generation cephalosporins is not an accurate proxy for ESBL production. It was also not surprising that strains KC-5, KC-10, KC-18, and KC-32, which harbored either *bla*_CTX-M_ or *bla*_SHV_, were susceptible to one or more third-generation cephalosporins, given that such genotype–phenotype discrepancy was previously described in clinical isolates [19]. Compared to our findings, a higher occurrence of ceftriaxone-resistant *E. coli* strains (17.3%) was reported in Mexico [29]. However, our findings contrast with a previous study that could not detect cefotaxime resistance in 319 *E. coli* strains isolated from raw milk and raw milk cheese in Brazil [27]. Interestingly, the relatively high frequency of third-generation cephalosporine-resistant *E. coli* strains detected in this study prompts increased attention of researchers to routinely perform susceptibility to this group of antibiotics in food isolates. We believe that testing food isolates for resistance to at least three third-generation cephalosporins, such as ceftriaxone, cefotaxime, and ceftazidime is critical for accumulating sufficient antimicrobial resistance data in this type of cheese and determining the burden of its consumption.

Globally, antimicrobial resistance genes in commensal bacteria are an indirect risk to public health, as they increase the genomic pool from which pathogenic bacteria can acquire resistance genes. In this study, resistance to quinolones, which was included in the WHO list of highest priority critically important antimicrobials used for human treatment [34], was detected in one strain (KC-53) (0.7%, 1/140) which carried the plasmid-mediated quinolone resistance gene, *qnrS* (Figure 2). On the other hand, PCR amplification of β-lactam resistance genes showed that 4.3% (6/140) and 2.8% (4/140) of isolates were positive for *bla*_SHV_, and *bla*_CTX-M_, respectively. Further, of the six tetracycline resistance genes screened, *tet*(A) and *tet*(B) were detected, with 2.1% (3/140) of strains carrying *tet*(A) and 0.71% (1/140) carrying *tet*(B). The antibiotic resistance genes detected in this study provided important clues about their ability to be horizontally transmitted, as they are usually carried on mobile genetic elements.

In Egypt, the over-counter availability of antibiotics and the lack of adequate antibiotic stewardship and laws to regulate the use of antibiotics for animals and people make it challenging to utilize antibiotics responsibly. As a result, finding multiple antibiotic-resistant and multidrug-resistant *E. coli* strains in retail raw milk cheese samples from Egypt is not surprising. However, in the absence of antimicrobial resistance pattern data for isolates from humans and animals in Egypt, it cannot be deduced with confidence whether contamination from animal, human, or both sources has occurred. Our findings corroborate that conducting national surveillance programs in Egypt for periodically monitoring the prevalence of antimicrobial-resistant *E. coli* strains in retail raw milk cheese is crucial to ensure its safety.

To gain insight into the role of karish as a vehicle for transmission of diarrheagenic *E. coli*, we examined different virulence factors in the isolated *E. coli* strains. One strain (KC-119) (0.71, 1/140) carried the *sxt*2 gene, which is alarming as epidemiological studies found that Stx2, not Stx1, is more frequently linked to severe illness and development of HUS [36,37]. Even though the absence of the intimin coding gene in this strain (*eae*) led to its exclusion of the enterohemorrhagic *E. coli* (EHEC) group but it still has the potential to be injurious to human health, considering that some *eae*-negative strains were found to be associated with human diseases [38]. Notably, the occurrence of STEC in raw milk cheeses showed a slightly wide variation between and within countries. For instance, on the one hand, the occurrence of STEC was 2.5%, 2.25%, 0%, and 0% in raw milk cheese samples from Italy, Egypt, the US, and Brazil, respectively [39,40,41]. On the other hand, relatively higher occurrences of 5.7% and 8% were reported in Switzerland and Mexico, respectively [29,42]. Several hypotheses have been proposed to explain this issue, including loss of *stx* genes during isolation, the presence of high levels of competing bacteria, and natural inhibitors produced within dairy products that interfere with STEC isolation [43]. Given the risk of STEC transmission to humans through raw milk cheese-eating, as well as the serious health consequences of STEC infection, further study is urgently needed to address this problem and develop an agreement on the best procedure for isolating STEC from raw milk cheeses.

A striking finding of our study is the detection of virulence pattern, *stx*1−, *stx*2+, *eae*−, that was frequently detected in STEC strains isolated from cattle in different countries all over the world, such as Argentina [38,44], France [45], and Portugal [46]. To increase our understanding of the spread of this pattern in STEC isolated from raw milk cheeses, we investigated its occurrence among other patterns in STEC strains isolated from natural milk cheeses worldwide in the last decade. To this end, we constructed a heat map for depicting the virulence patterns of STEC reported in 17 studies. As illustrated in Figure 3, the virulence patterns were categorized into seven distinct clusters representing seven virulence patterns. Eleven studies (61.11%, 11/18), including our study, found the virulence pattern “*stx*1−, *stx*2+, *eae*−,” which makes up cluster 1. STEC was shown to carry *stx*2 with one or both *stx*1 and *eae* genes in 13 (72.2%, 13/18) investigations, forming clusters 4, 5, and 7. Our strain clustered with strains isolated from raw milk cheeses in Egypt and European countries, including France, England, Italy, and Switzerland. Surprisingly, this virulence pattern was detected in 27 out of 29 STEC strains isolated from raw milk cheese in Switzerland [42]. We assume that STEC strains carrying this virulence pattern may find their way to cheese through fecal contamination from cattle during the milking process. Worryingly, given that cheese-making processes can induce phage production, leading to the presence of Stx phages as free particles in cheeses [47], the possibility of other pathogenic and non-pathogenic bacteria acquiring *stx* genes remains. Our findings, together with the data depicted in Figure 3 from the globally disseminated STEC, reveal that raw milk cheese is a potential reservoir of *stx* genes, particularly *stx*2, posing a threat to human health.

## 4. Conclusions

The high prevalence and loads of coliforms and fecal coliforms found in this study indicated that hygienic-sanitary failures occurred throughout the production process of most retail karish cheese in Egypt. In line with the earlier idea of employing indicators to track antimicrobial resistance in humans and food-producing animals [15], we propose that antimicrobial resistance surveillance systems in raw milk cheese be harmonized using markers like third-generation cephalosporin-resistant *E. coli*. These indicators are recommended to reflect the current state of antimicrobial resistance in this type of cheese and, as a result, to identify potential health hazards associated with its consumption. On the other hand, considering that contamination of raw milk cheese by STEC may not be entirely controlled through the cheesemaking process [59,60], the finding of *stx*2-positive *E. coli* in this study, as well as other prior studies, highlights the importance of routine STEC surveillance in retail raw milk cheese to assure its safety.

## Figures and Tables

**Figure 1 foods-11-00332-f001:**
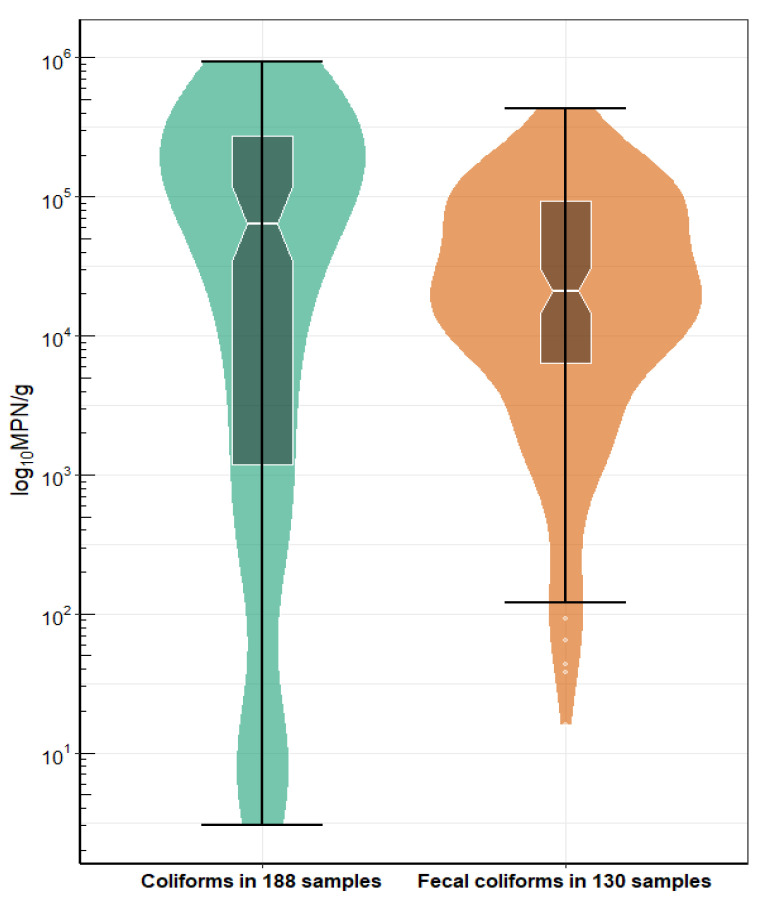
Violin plots overlaid on box and whisker plots showing the distribution of log 10 MPN/g of coliforms and fecal coliforms in raw milk cheese samples. The box represents 50% of data values, and the horizontal line within the box signifies the median. The upper and lower boundaries of the box are the 75th and 25th percentile, respectively. The whiskers (error bars) above and below the box represent the highest and lowest values within 1.5 times the interquartile range and indicate the 90th percentiles and 10th percentiles, respectively. White spots represent the outliers. The width of the violin plot reveals the frequency of samples.

**Figure 2 foods-11-00332-f002:**
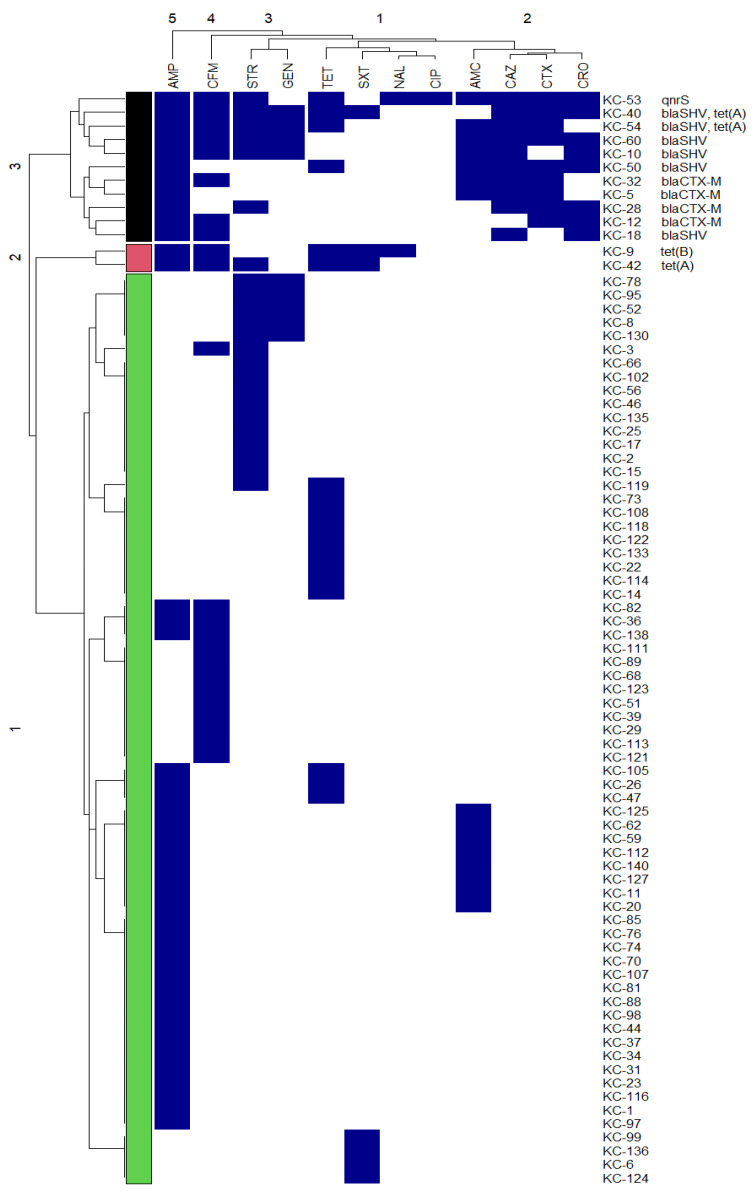
Heatmap showing antibiotic resistance profiles of thermotolerant *E. coli* strains. The numbers of strains are on the right of the heat map, followed by the identified antibiotic resistance genes. The dark blue color of the box indicates the resistance to the antimicrobial, while the white color indicates the susceptibility to the antimicrobial. The dendrogram on the left reflects the hierarchical clustering of antibiotic resistance profiles in the isolated strains. The numbers on the dendrogram (1 to 3) indicate the numbers of clusters. The dendrogram on the top reflects the hierarchical clustering of screened antibiotics. Abbreviations of the antimicrobials are AMC, amoxicillin-clavulanic acid; AMP, ampicillin; CAZ, ceftazidime; CFM, cefixime; CIP, ciprofloxacin; CRO, ceftriaxone; CTX, cefotaxime; GEN, gentamicin; NAL, nalidixic acid; STR, streptomycin; SXT, sulfamethoxazole-trimethoprim and TET, tetracycline.

**Figure 3 foods-11-00332-f003:**
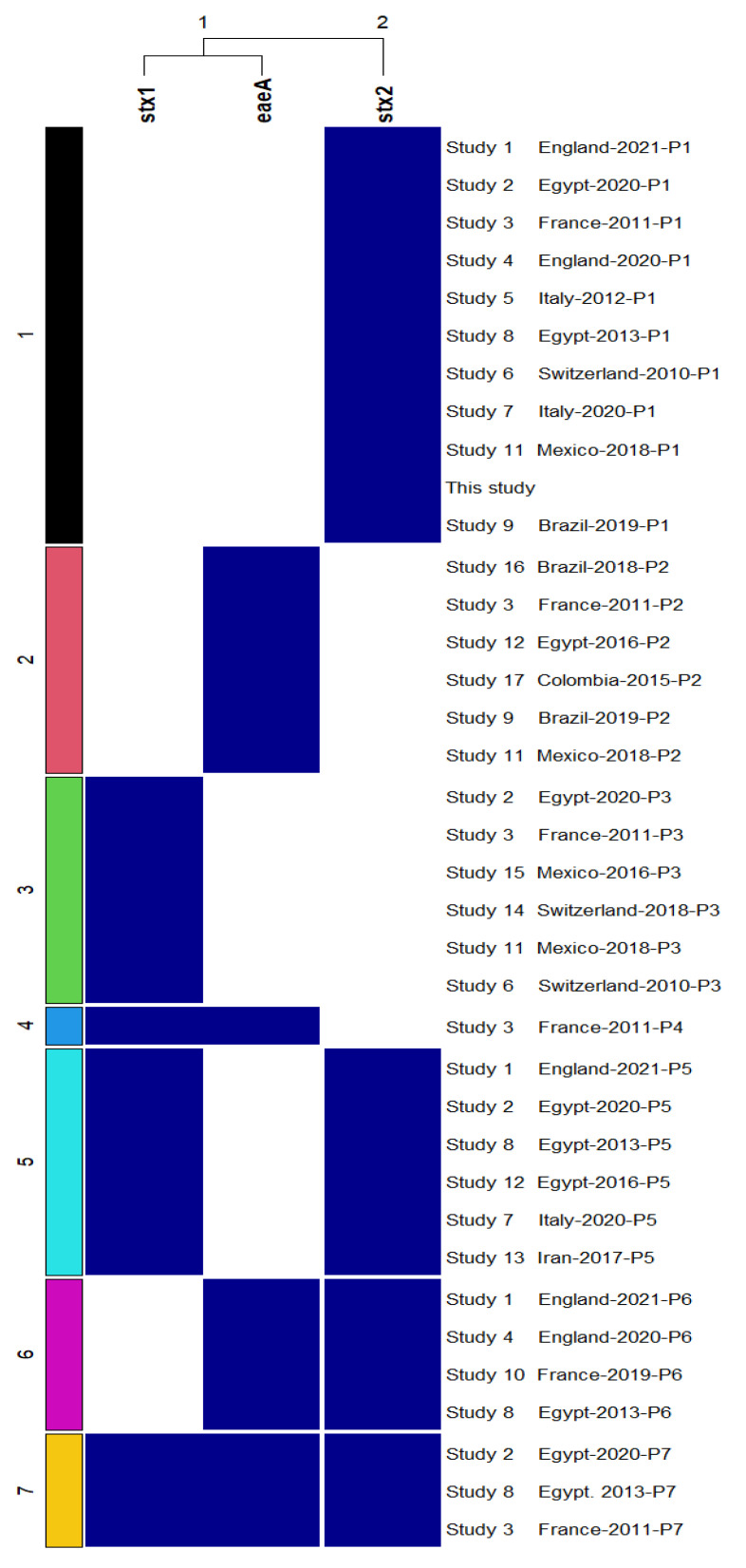
Heatmap showing virulence profiles of our strain and published strains isolated from raw milk cheeses in different countries in the last decade. All virulence patterns discovered in each of the 17 investigations were depicted once, regardless of their frequency. The dark blue color of the box indicates the presence of the gene, while the white color indicates the absence of the gene. On the right side of the heatmap, the country of origin of each strain and the year of publication of each study were written. The following studies were included: Study 1 [48]; Study 2 [49]; Study 3 [50]; Study 4 [51]; Study 5 [39]; Study 6 [42]; Study 7 [52]; Study 8 [53]; Study 9 [27]; Study 10 [54]; Study 11 [29]; Study 12 [40]; Study 13 [55]; Study 14 [56]; Study 15 [57]; Study 16 [41]; and Study 17 [58].

## Data Availability

The datasets used and analyzed during the current study are available from the corresponding author on reasonable request.

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
