# Peer review of "Loads of Coliforms and Fecal Coliforms and Characterization of Thermotolerant Escherichia coli in Fresh Raw Milk Cheese"

_foods, 2022, doi:10.3390/foods11030332_

Round 1

Reviewer 1 Report

COMMENTS.

This study examines the presence of thermotolerant Escherichia coli in raw milk cheese and the outcomes associated to its presence. The findings are interesting and could represent a suitable method for evaluate the hygienic status of karish.  The manuscript contains some flaws that requires deep revisions.

ABSTRACT.

Text does not contain a brief description of the bacteria and what its presence may mean in raw milk cheese.

INTRODUCTION.

Line 38-39. Please indicate other possible microbiological contaminations which could occur during the manufacturing process. Moreover, add some references about the above contaminations.

Line 56. Replace “categorized based” with “grouped according to”.

Line 64. Remove “of”.

Line 67. Replace “from” with “associated to”.

Lines 68-70. Please reorganize the sentence.

Lines 82-86. Please reorganize the sentence.

Lines 87-91. The sentence is too long. Please revise and reorganize.

Lines 91-94. Please you should better describe why did you have carried out this research.

Please add more recent references. Most of the references used for this research were published more than ten years ago.

MATERIALS AND METHODS.

Lines 105-109. Please add reference for this method.

Lines 115-122. Why did you use only these antibiotics? Are these reported in the “Clinical and Laboratory Standards Institute guideline”. If there are other antibiotics in the above guideline, you should explain why did you choose only these.  

Lines 123-134. Please add reference for ESBL production.

RESULTS AND DISCUSSION.

Lines 178-180. This sentence could be removed and added to the introduction section and, please, add the Regulation about the Coliforms count and limit in karish cheese.

Lines 188-189. What is 150? Are the strains isolated from karish sample? Please clarify this data.

Line 190. In this case, E. coli is thermotolerant or not?

Lines 195-198. Why did you introduce this data?  

Lines 213-216: Please clarify the 70, 46 and 20 isolates. You should give these number as n. isolates/ n. totality and explained also the %, as you have described at line 189, line 208 or lines 216. Moreover, rephrase the sentence to better describe these data. If all these data are referred to 140 strains, one is absent from the description.

Line 306. Please control the reference 36 and 51. They are not referred to Italy.

Please add more recent references. Most of the references used for this research were published more than ten years ago.

Author Response

Comment: This study examines the presence of thermotolerant Escherichia coli in raw milk cheese and the outcomes associated to its presence. The findings are interesting and could represent a suitable method for evaluate the hygienic status of karish.

Reply: We would like to thank you much for all your valuable comments that helped in improving our manuscript.

ABSTRACT.

Comment: Text does not contain a brief description of the bacteria and what its presence may mean in raw milk cheese.

Reply: We agree with you but according to the instructions to the author of this journal the abstract should not exceed 200 words. Now it is 197 words and contained a brief description of our results.

INTRODUCTION.

Comment: Line 38-39. Please indicate other possible microbiological contaminations which could occur during the manufacturing process. Moreover, add some references about the above contaminations.

Reply: Other possible microbiological contaminations were added, and 3 recent references were also added.

Comment: Line 56. Replace “categorized based” with “grouped according to”.

Reply: “categorized based” was replaced by “grouped according to”.

Comment: Line 64. Remove “of”.

Reply: “of” was removed.

Comment: Line 67. Replace “from” with “associated to”.

Reply: “from” was replaced by “associated to”

Comment: Lines 68-70. Please reorganize the sentence.

Reply: The sentence was reorganized.

Comment: Lines 82-86. Please reorganize the sentence.

Reply: The sentence was reorganized.

Comment: Lines 87-91. The sentence is too long. Please revise and reorganize.

Reply: The sentence was shortened and reorganized.

Comment: Lines 91-94. Please you should better describe why did you have carried out this research.

Reply: This sentence was modified in the revised manuscript to describe why did we have carried out this research. We removed the part of this sentence that describes the methods by which we have performed this research.

Comment: Please add more recent references. Most of the references used for this research were published more than ten years ago.

Reply: We updated the references and added some new references. The revised manuscript contains 46 references (66.6%, 46/69) published after 2015. Regarding the most recent references, our revised manuscript contained 19 references (27.5%, 19/69) which published in the last 2 years, 2020 and 2021.

MATERIALS AND METHODS.

Comment: Lines 105-109. Please add reference for this method.

Reply: A reference was added.

Comment: Lines 115-122. Why did you use only these antibiotics? Are these reported in the “Clinical and Laboratory Standards Institute guideline”. If there are other antibiotics in the above guideline, you should explain why did you choose only these.

Reply: We added the following sentence: “The selected antimicrobials are important in both clinical and agricultural settings and utilized to assess acquired resistance in E. coli and other Enterobacteriaceae”. Further, in the results and discussion section, we shed the light on their importance and wrote “Interestingly, in the recent ranking of medically essential antimicrobials conducted by the World Health Organization (WHO), third-generation cephalosporins were classified as "highest priority critically important antimicrobials"”

Comment: Lines 123-134. Please add reference for ESBL production.

Reply: A reference was added.

RESULTS AND DISCUSSION.

Comment: Lines 178-180. This sentence could be removed and added to the introduction section and, please, add the Regulation about the Coliforms count and limit in karish cheese.

Reply: The sentence was transferred to the introduction and a reference of the Egyptian regulation was added.

Comment: Lines 188-189. What is 150? Are the strains isolated from karish sample? Please clarify this data.

Reply: A sentence was added to clarify this data.

Comment: Line 190. In this case, E. coli is thermotolerant or not?

Reply: Yes, E. coli is thermotolerant and the sentence was modified as “The high occurrence of thermotolerant E. coli in our samples revealed possible fecal contamination during milk collection and cheese processing”

Comment: Lines 195-198. Why did you introduce this data?

Reply: This data supports a statement mentioned in a preceding paragraph. We wrote that “Our data reveals how unsanitary manufacturing practices during production and handling of raw milk cheese may jeopardize the product's microbiological quality, the situation that was reported in both developed and developing countries”. Therefore, we preferred to compare the occurrence of E. coli in our study with literatures published in both developed and developing countries.  

Comment: Lines 213-216: Please clarify the 70, 46 and 20 isolates. You should give these number as n. isolates/ n. totality and explained also the %, as you have described at line 189, line 208 or lines 216. Moreover, rephrase the sentence to better describe these data. If all these data are referred to 140 strains, one is absent from the description.

Reply: This error in typing was fixed and the sentence was rephrased in the revised manuscript. All percentages were added in the revised manuscript.

Comment: Line 306. Please control the reference 36 and 51. They are not referred to Italy.

Reply: The reference was corrected in the revised manuscript (Switzerland instead of Italy).

Comment: Please add more recent references. Most of the references used for this research were published more than ten years ago.

Reply: We updated the references and added some new references. The revised manuscript contains 46 references (66.6%, 46/69) published after 2015. Regarding the most recent references, our revised manuscript contained 19 references (27.5%, 19/69) which published in the last 2 years, 2020 and 2021.

Reviewer 2 Report

Overall, the article is interesting and correctly written. However, it requires some supplementation. Detailed comments are included in the attached pdf file. 

Author Response

Comment: Overall, the article is interesting and correctly written. However, it requires some supplementation. Detailed comments are included in the attached pdf file.

Reply: We would like to thank you much for all your valuable comments that helped in improving our manuscript.

Comment: Please add when the samples were tested since their arrival to the laboratory. The conditions under which they were stored before testing should be added.

Reply: This data was provided in the revised manuscript.

Comment: On what standard was the first dilution in peptone and not sodium citrate made?

Reply: Sodium citrate 2% solution at 45°C is proposed as an ideal diluent for cheese samples as it facilitates the dispersion of curd and consequent release of microorganisms. However, Bird et al. (2020) published a recent paper (Journal of AOAC International, 103(2), 2020, 513–522) entitled “Evaluation of the 3MTM PetrifilmTM Rapid E. coli/ Coliform Count Plate for the Enumeration of E. coli and Coliforms: Collaborative Study, First Action: 2018.13” and stated that “Do not use diluents containing citrate, bisulfate or thiosulfate; they can inhibit growth”. Even though we did not use this method, but we preferred to exclude the likelihood of inhibiting growth of coliforms using sodium citrate as the first diluent. Moreover, in our case, karish cheese is a soft fresh cheese that can be easily dispersed by homogenization in a stomacher in any diluent.

Comment: add source of all reagents (city and country)

Reply: The city and country of all reagents were added.

Comment: all abbreviations used for the first time should be explained

Reply: Escherichia coli Broth (EC broth) was added in the manuscript.

Comment: Please fill in what software was used and its source.

Reply: The software and its source were added in the revised manuscript.

Comment: please update the literature, as only 15 of the 69 items are publications from the last 5 years.

Reply: We updated the references and added some new references. The revised manuscript contains 46 references (66.6%, 46/69) published after 2015. Regarding the most recent references, our revised manuscript contained 19 references (27.5%, 19/69) which published in the last 2 years, 2020 and 2021.

Reviewer 3 Report

The main question addressed in the manuscript „Loads of Coliforms and Fecal Coliforms and Characterization of Thermotolerant Escherichia coli in Fresh Raw Milk Cheese” is to assess the hygienic status of raw milk cheese in Egypt by quantifying loads of coliforms and fecal coliforms and determining the trends of virulence and antimicrobial resistance in thermotolerant E. coli.

At the first part of their work Authors collected two hundred samples of karish (Egyptian fresh raw milk cheese) and then performed phenotypic and molecular characteristics of coliforms including antibiotic resistance genes analysis.

The paper is technically sound and undertaken research topic is particularly interesting in food microbiology. In the Manuscript experiments are well designed. The methods are appropriate and properly conducted and results are valuable and could contribute to the development of public health protection monitoring and control activities for dairy industries. The claims are fully supported by the experimental data and appropriately discussed in the context of previous literature. Conclusion is consistent with the evidence and arguments presented. The manuscript is clearly written.

Author Response

Comments:

The main question addressed in the manuscript „Loads of Coliforms and Fecal Coliforms and Characterization of Thermotolerant Escherichia coli in Fresh Raw Milk Cheese” is to assess the hygienic status of raw milk cheese in Egypt by quantifying loads of coliforms and fecal coliforms and determining the trends of virulence and antimicrobial resistance in thermotolerant E. coli.

At the first part of their work Authors collected two hundred samples of karish (Egyptian fresh raw milk cheese) and then performed phenotypic and molecular characteristics of coliforms including antibiotic resistance genes analysis.

The paper is technically sound and undertaken research topic is particularly interesting in food microbiology. In the Manuscript experiments are well designed. The methods are appropriate and properly conducted and results are valuable and could contribute to the development of public health protection monitoring and control activities for dairy industries. The claims are fully supported by the experimental data and appropriately discussed in the context of previous literature. Conclusion is consistent with the evidence and arguments presented. The manuscript is clearly written.

Reply: We would like to thank you much for all your valuable comments.

Reviewer 4 Report

Dear all,

below are my comments and suggestions

Manuscript ID: foods-1543838

The research entitled " Loads of Coliforms and Fecal Coliforms and Characterization of Thermotolerant Escherichia coli in Fresh Raw Milk Cheese " is interesting and useful research to provide information on  the hygienic status of raw milk cheese and determining the trends of virulence and antimicrobial resistance in thermotolerant Escherichia coli

However, some changes need to be made in the manuscript itself.

Introduction: The introduction can be improved and include more relevant references

Materials and Methods. It was difficult because the text is not understandable in some places. Please verify the method description.

Please indicate the moment in the period of shelf life when the samples were collected and the storage conditions expected in the intended market.

The number of bibliographic sources is adequate, but less than 35% of the total bibliographic sources are from the last 5  years.

There are some grammatical errors and instances of badly worded/constructed sentences throughout the manuscript. Please refine the language carefully. 

Author Response

Comment: The research entitled " Loads of Coliforms and Fecal Coliforms and Characterization of Thermotolerant Escherichia coli in Fresh Raw Milk Cheese " is interesting and useful research to provide information on the hygienic status of raw milk cheese and determining the trends of virulence and antimicrobial resistance in thermotolerant Escherichia coli

Reply: We would like to thank you much for all your valuable comments that helped in improving our manuscript.

Introduction:

Comment: The introduction can be improved and include more relevant references.

Reply: We improved the introduction and added some relevant references in the revised manuscript.

Comment: Materials and Methods. It was difficult because the text is not understandable in some places. Please verify the method description.

Reply: Some sentences in the material and methods section were reorganized.

Comment: Please indicate the moment in the period of shelf life when the samples were collected and the storage conditions expected in the intended market.

Reply: All this data was added in the revised manuscript.

Comment: The number of bibliographic sources is adequate, but less than 35% of the total bibliographic sources are from the last 5 years.

Reply: We updated the references and added some new references. The revised manuscript contains 46 references (66.6%, 46/69) published after 2015. Regarding the most recent references, our revised manuscript contained 19 references (27.5%, 19/69) which published in the last 2 years, 2020 and 2021.

Comment: There are some grammatical errors and instances of badly worded/constructed sentences throughout the manuscript. Please refine the language carefully.

Reply: The grammatical errors were corrected.

Reviewer 5 Report

Dear authors/editor

I had reviewed the article “Loads of coliforms and fecal coliforms and characterization of thermotolerant Escherichia coli in fresh raw milk cheese.” This work aimed to study the hygienic status of raw milk cheese in Egypt by quantifying loads of coliforms and determining the trends of virulence and antimicrobial resistance. In general, the manuscript is well written, and it is easy to follow. In addition, the authors described clearly their materials and methods used. I only have some questions about their results and the experiments performed.

Bacterial growth depends on the intrinsic characteristics of foods. However, you don´t describe any inherent feature of this raw milk cheese. For instance, which was the average pH of cheese samples? or the average water content (or water activity)? or the salt content in the cheese samples? These parameters are essential to understand the growth of thermotolerant Escherichia coli in the cheese samples.

Referring to the conclusion, the authors described Karish cheese as an acid-induced coagulated cheese similar to quark and cottage cheeses. Therefore, I suppose that milk is fermented using the “natural” microflora found in the raw milk (please include some information in the introduction). As the authors described in their cheese samples, Escherichia coli, and many other bacteria may grow to very high numbers in this fermentation process. However, the fecal coliforms also could be incorporated after cheese making and during its manipulation and storage. Please include some information or your thoughts about this question in your manuscript.

Author Response

Comments: I had reviewed the article “Loads of coliforms and fecal coliforms and characterization of thermotolerant Escherichia coli in fresh raw milk cheese.” This work aimed to study the hygienic status of raw milk cheese in Egypt by quantifying loads of coliforms and determining the trends of virulence and antimicrobial resistance. In general, the manuscript is well written, and it is easy to follow. In addition, the authors described clearly their materials and methods used. I only have some questions about their results and the experiments performed.

Reply: We would like to thank you much for all your valuable comments that helped in improving our manuscript.

Comment: Bacterial growth depends on the intrinsic characteristics of foods. However, you don´t describe any inherent feature of this raw milk cheese. For instance, which was the average pH of cheese samples? or the average water content (or water activity)? or the salt content in the cheese samples? These parameters are essential to understand the growth of thermotolerant Escherichia coli in the cheese samples.

Reply: We agree with you that this data is important, but it was away from the scope of this study. Establishing a correlation between the intrinsic parameters of cheese and the level of contamination with coliforms, fecal coliforms, and E. coli needs an independent study. We plan to conduct this research in the future.

Comment: Referring to the conclusion, the authors described Karish cheese as an acid-induced coagulated cheese similar to quark and cottage cheeses. Therefore, I suppose that milk is fermented using the “natural” microflora found in the raw milk (please include some information in the introduction). As the authors described in their cheese samples, Escherichia coli, and many other bacteria may grow to very high numbers in this fermentation process. However, the fecal coliforms also could be incorporated after cheese making and during its manipulation and storage. Please include some information or your thoughts about this question in your manuscript.

Reply: In response to your comment, we added in the introduction the following sentence: “However, lactic acid is created when lactose is fermented by the endogenous microflora of milk during the cheesemaking process which is potent against some bacteria”. More information about sources of contamination were added in the introduction supported by references.

Round 2

Reviewer 2 Report

After corrections manuscript can be accepted.

Reviewer 5 Report

The authors have answered all my questions and attended all my queries.